# Best Practices for the Use of High-Frequency Ultrasound to Guide Aesthetic Filler Injections—Part 1: Upper Third of the Face

**DOI:** 10.3390/diagnostics14161718

**Published:** 2024-08-08

**Authors:** Rosa Sigrist, Stella Desyatnikova, Maria Cristina Chammas, Roberta Vasconcelos-Berg

**Affiliations:** 1Department of Radiology, School of Medicine, University of São Paulo, São Paulo 05403-010, Brazil; mcchammas@hotmail.com; 2The Stella Center for Facial Plastic Surgery, Seattle, WA 98101, USA; stella@doctorstella.com; 3Margarethenklinik-University Hospital of Basel, CH-4051 Basel, Switzerland; roberta.vasconcelos-berg@usb.ch

**Keywords:** filler, hyaluronic acid, guided injections, forehead, temple, glabella, Doppler ultrasound

## Abstract

Filler injections in the upper face pose significant challenges due to its complex anatomy and proximity to vascular structures. High-frequency Doppler ultrasound offers real-time visualization of facial anatomy, improving both safety and aesthetic outcomes. This paper presents a detailed overview of the ultrasonographic anatomy of the temples, forehead, and glabella, along with reproducible, ultrasound-guided filler injection techniques for these areas. We use two scanning techniques previously described: “scan before injecting” and “scan while injecting”, applicable to subdermal, interfascial, and supraperiosteal planes in the temporal region, as well as the glabella, forehead, and supraorbital region. Ultrasound guidance for filler injections in the upper face can enhance procedural efficacy and safety. By integrating real-time imaging, practitioners can navigate the intricate vascular anatomy more effectively, thereby minimizing the risk of complications. This study highlights the need for ongoing research and continuous education to further refine these techniques and improve patient outcomes.

## 1. Introduction

Filler injections in the upper face, including the temples, forehead, and glabella, pose challenges due to its complex anatomy and proximity to vascular structures. The rich vascularity of the temporal region and the convergence of vessels’ paths from the forehead with the region of glabellar wrinkles must be carefully considered during treatment [1]. Additionally, the proximity to the eye, coupled with the frequent anastomoses between the external and internal carotid systems in this area, renders the upper third of the face a high-risk region for secondary effects of vascular embolization, such as blindness [2] and cerebral infarction [3].

High-frequency ultrasound is a radiation-free imaging technique that can assist in real-time visualization of facial anatomical structures, ensuring the desired plan is achieved [4]. The use of Doppler ultrasound (DUS) further highlights vascular structures and their positioning in relation to the cannula or needle, which can enhance the safety of aesthetic injections [4,5].

Ultrasound-guided temporal filling techniques have been described by Desyatnikova [6], Kadouch et al. [7], Lee et al. [8], Kim et al. [9], and Surek [10]. All of them acknowledge the complexity of the anatomical layers in this area and describe methods for accessing the desired plane of injection with ultrasound assistance.

Frontal and glabellar region ultrasound anatomical features have been addressed by Desyatnikova [11], Cotofana et al. [12], Tansatit et al. [13], Bravo et al. [14], and Choi et al. [15]. However, these articles do not propose a method for guiding injectable procedures. Phumyoo et al. [16], in addition to describing the anatomical characteristics of this area, proposed that the main arteries of this region should be identified by ultrasound prior to hyaluronic acid injection.

The aim of this article is to offer a comprehensive overview of the ultrasonographic anatomy of the upper third of the face and to provide detailed and reproducible ultrasound-guided techniques in the temporal, frontal, and glabellar regions.

## 2. Materials and Methods

This narrative review is based on the authors’ experience performing high-frequency DUS-guided injections in the upper face. These techniques and the recommendations provided in this paper represent solely the authors’ viewpoint and should be treated accordingly.

The ultrasound systems used were a LOGIQ E10 (GE Healthcare, Waukesha, WI, USA) with a linear probe ranging from 6 to 24 MHz, a Venue Fit (GE Healthcare, Waukesha, WI) with a linear probe ranging from 4 to 20 MHz, and an ACUSON Sequoia (Siemens Medical Solutions, Mountain View, CA, USA), with a linear probe ranging from 6 to 18 MHz.

Before the procedures, the treatment area was disinfected, and the linear probe was covered with a sterile transparent dressing, Opsite^®^ (Smith & Nephew Medical, Suzhou, China), following a technique previously described by our group [5].

The injections were performed either with a blunt 22 G or 25 G cannula, or a 27 G or 30 G needle.

As previously suggested by this group [5], we categorize the methods of guided injection into the following didactic approaches:**Scan before injecting:** The treatment area is scanned immediately before injection to assess the presence of vessels. The trajectory of the main arteries might be marked on the patient’s skin.**Scan while injecting:** In this case, the cannula is visualized in real time and positioned in the desired anatomical region, avoiding vascular structures.

As far as the position between the US probe and the cannula is concerned, in-plane and out-of-plane techniques can be used. In-plane technique is preferred as one can see in real-time the long axes of the cannula. However, if the cannula is too close to the vessel, the direction of the probe can be changed to out-of-plane technique to confirm that the cannula is not inside the vessel. 

## 3. Results

### 3.1. Temporal Region

#### 3.1.1. Sonographic Anatomy

The temporal region is an anatomically complex area with the following layers visualized on ultrasound (from superficial to deep) [6,9] (Figure 1):
Epidermis: hyperechoic line.Upper dermis: hypoechoic homogeneous layer.Lower dermis: usually more echogenic layer than the upper dermis.Subcutaneous fat tissue: a hypoechoic layer composed of fat lobules and hyperechoic septae.Superficial temporal fascia/Superficial musculoaponeurotic system (SMAS): linear hyperechoic layers enveloping the superficial temporal artery and vein.Sub-SMAS fat: also called innominate fascia, it is a hypoechoic layer composed of loose connective tissue and fat lobules. This is the interfascial plane.Superficial lamina of the deep temporal fascia: a hyperechoic line, which is juxtaposed to the intermediate temporal fat compartment.Intermediate fat compartment (loose areolar tissue): a hypoechoic triangular layer composed of fat lobules and hyperechoic septae. The middle temporal vein can be encountered in this layer.Deep lamina of the deep temporal fascia: a hyperechoic line, which is deep to the intermediate temporal fat compartment.Temporal muscle: a large hypoechoic structure above the bone, where the anterior and posterior deep temporal arteries are located.Temporal extension of the buccal fat compartment: a hypoechoic fat compartment adjacent to the lateral orbital rim, connected to the buccal fat pat. Easily recognized when patient is asked to open and close the mouth, this fat pad can be found under the deep lamina of the deep temporal fascia.Bone: a thin hyperechoic line with acoustic shadowing.

The main vessels in the temple are the frontal and parietal branches of the superficial temporal artery, which run between the lamina of the superficial temporal fascia, the anterior and posterior deep temporal arteries located deep in the temporal muscle, the zygomaticotemporal artery, the sentinel vein, and the middle temporal vein running in the temporal fat pad, between the deep and superficial laminae of the deep temporal fascia [17]. 

#### 3.1.2. Ultrasound-Guided Filling Techniques of the Temporal Region

The volumization of the temporal region aims primarily to visually restore lost volume in the temporal region due to the natural aging process; however, it can also occur due to weight loss or chronic diseases. Additionally, elevation of anatomical structures in the temporal region by filler can lead to a lifting effect, raising the structures of the lateral face, especially the eyebrow and upper eyelid. Multiple techniques [18] have been described for temporal volumization, some of which address adjacent areas, such as the region posterior to the hairline and the preauricular region. Our article will focus on three main techniques, most commonly used by injectors.

Technique 1: Subdermal filler placement

The primary goal of this technique is to efficiently volumize the temporal region. This injection method positions the product in the subcutaneous fatty tissue of the anterior temple, which is limited posteriorly by the hairline. The entry point on the skin is situated in the posterior portion of the zygomatic arch and ideally utilizes a 22 G 50 mm blunt-tip cannula.

As the temporal region is richly vascularized, mapping all vessels with ultrasound before injection would be a labor-intensive task, significantly increasing treatment time without necessarily adding safety to the procedure. 

For this injection method the main vascular structures to be avoided are the superficial temporal artery and vein, which lie within the laminae of the superficial temporal fascia. Since this plane favors the use of a cannula, we employ the “scan while injecting” technique. 

The initial step involves inserting the cannula into the skin and subcutaneous tissue and positioning it in the temple area superficially. At this stage, ultrasound is not employed. Subsequently, the probe is positioned along the cannula’s trajectory using a small amount of sterile gel for conductivity (Figure 2a). On the screen, an observation is made to ensure that the cannula is correctly positioned within the desired layer and its tip is not in contact with the superficial temporal artery or vein branches (Figure 2b).

Technique 2: Interfascial filler placement

The goal of this technique is to volumize the temporal region as well. This injection method positions the product in the interfascial layer, between the superficial temporal fascia and the superficial layer of the deep temporal fascia. It is a potential space with a small amount of fat and loose connective tissue, but it can expand significantly upon filler placement. The entry point on the skin is situated in the posterior third of the zygomatic arch, since the temporal branch of the facial nerve typically crosses the zygomatic arch in the middle. A 22 G 50 mm blunt-tip cannula is used.

Using ultrasound-guided injection for this layer does not require pre-injection vascular mapping, since we can visualize the vessels during the injection in real time. The vascular structures to be avoided are the superficial temporal artery and vein, middle temporal vein, and sentinel vein.

The initial step involves creating a pilot hole with a needle and inserting the cannula through the skin, subcutaneous tissue, and superficial temporal fascia down to the level of deep temporal fascia and sliding the cannula in the cephalic direction along the surface of the deep temporal fascia. Subsequently, the probe is positioned along the cannula’s trajectory using a small amount of sterile gel for conductivity (Figure 3). On the screen, we can visualize the cannula between the fascial layers in the small interfascial space, confirming cannula tip position away from the superficial temporal artery and vein, sentinel vein, and middle temporal vein.

A small amount of filler is injected, and expansion of this layer is observed in real time, confirming proper position of the cannula. Retrograde linear threads of filler are then placed. Multiple threads can be placed in a fanning fashion, while being observed on the ultrasound screen in real time, gradually expanding the volume of the interfascial space as necessary. Gentle pressure may be necessary to advance the cannula through the inferior temporal septum. Anteriorly, the filling is limited by the fascial fusion of the lateral orbital thickening.

Technique 3: Supraperiosteal filler placement

This technique involves placing the filler in the supraperiosteal plane of the anterior temple, inserting a needle perpendicularly to the skin. The injection point for the technique classically called “one up, one over” is located 1 cm lateral to the temporal crest, and 1 cm cranial to the orbital rim. This approach is recommended for severe volume loss of the superior anterior temple, which results in increased visibility of the temporal crest. Additionally, elevation of temporal tissues allows the tail of the eyebrow to be lifted by this technique.

Initially, the injection point should be marked in the temporal region at the location previously described (Figure 4a). For this point, we use the “scan before injecting” technique (Figure 4b) and scan the area in two perpendicular planes to rule out the presence of the anterior deep temporal artery (Figure 4c). If the artery is found at this point (Appendix A), the injection point should be moved. Another nearby point can be marked, repeating the same procedure to ensure that the artery’s path does not coincide with the area to be treated.

Once the injection point is decided, a 27 G sharp-tip needle is inserted perpendicular to the skin surface (Figure 4d) until bony contact is established without ultrasound guidance, since being perpendicular the needle cannot be visualized on ultrasound. Even though the region has been scanned previously, a prolonged aspiration is performed before injection due to the high-risk of this area, and then the product is slowly applied.

### 3.2. Frontal Region (Glabella, Forehead, and Supraorbital Region)

#### 3.2.1. Sonographic Anatomy

The layers of the forehead can be assessed sonographically [11] from superficial to deep (Figure 5a,b):
Epidermis: a hyperechoic line.Upper dermis: a hypoechoic homogeneous layer.Lower dermis: a hyperechoic layer.Subcutaneous fat tissue: a hypoechoic layer composed of fat lobules and hyperechoic septae.Suprafrontalis fascia: a thin hyperechoic upper layer of galea aponeurotica.Frontalis muscle: a hypoechoic band-like structure.Retro-Orbicularis Oculi Fat compartment (ROOF): a hyperechoic fibrous fat layer that separates the frontalis muscle from the bone. It can be appreciated in the inferolateral part of the forehead.Periosteum and subfrontalis fascia: a hyperechoic line showing combined imaging of these structures, with acoustic shadowing below. Due to the convexity and reflective nature of the frontal bone, there is usually a mirror imaging artefact.

As ultrasound is a dynamic imaging technique, it is also possible to evaluate the contraction of the frontalis in the forehead (Figure 6a,b). The contraction of the frontal muscle occurs uniformly, causing the juxtaposed subcutaneous fat tissue to appear condensed as the frontal lines (wrinkles) are formed (Appendix A).

The main arteries of the frontal region are the superficial and deep branches of the supratrochlear and supraorbital arteries, originating from the ophthalmic artery, a branch of the internal carotid artery [19]. Anastomosis with the frontal branches of the superficial temporal arteries may be present.

The supratrochlear and supraorbital arteries originate deep, emerging, respectively, from the supratrochlear and supraorbital foramina. However, proceeding cephalically, they become more superficial [12].

The glabella is additionally vascularized by the central and paracentral arteries [13,20]. In this area, there are the procerus and corrugator supercilii muscles. The procerus muscle, like the frontalis, is a hypoechoic band-like structure, while the corrugator supercilii muscle is a fan-like hypoechoic structure (Figure 7).

#### 3.2.2. Ultrasound-Guided Filler Injection Techniques for the Frontal Region

We recommend that filler procedures in this area be performed only by experienced injectors using ultrasound guidance. Additionally, to minimize the risk of occlusion, we recommend using blunt cannulas with a diameter of 25 G or larger. To further enhance the safety of the procedure, we combine both techniques in this region:

Scan before injecting

First, the glabella, forehead, and supraorbital regions are scanned, and the supratrochlear and supraorbital arteries are marked on the skin (Figure 8). If the central and paracentral arteries are present medially, they are marked on the skin. Since these vessels are small in caliber, it can be difficult to locate them in the longitudinal plane. In these cases, we recommend scanning perpendicular to the vessel, adjusting the Doppler scale and gain, marking sequential points on the skin corresponding to the cross-section of the vessel, and linking them to form a continuous line. When scanning along the longitudinal axis of the vessel, the color box should be steered in the direction of the vessel to maximize the color Doppler adjustment.

Scan while injecting

After marking the arterial path, the entry point is defined. The cannula is inserted, and the transducer is used again to ensure that the cannula is outside of the vessels and in the correct plane (Figure 9). If the path of one or both supratrochlear arteries coincides with the glabellar wrinkle, ultrasound should be used to confirm that the cannula is not at the same depth as the artery. The injection must be performed with extreme caution, using low-elasticity (low G′) hyaluronic acid and a small amount of product to avoid external compression.

#### Glabella

In the glabella, the primary area of interest for filler injections are the vertical wrinkles (“frown lines”) caused by the repetitive action of the corrugator muscles. According to Cotofana et al. [21], the mean distance between the supratrochlear artery and the ipsilateral vertical glabellar frown line at rest was 10.59 (4.0) mm (range, 2.9–19.0 mm) in males and 8.21 (4.0) mm (range, –3.3–14.2 mm) in females. Due to this proximity to glabellar vessels, the glabella is considered one of the highest-risk areas of the face, with filler injections in this region accounting for 27.1% of blindness cases reported in the literature in 2019 [22]. Therefore, the use of DUS in the glabella elucidates the trajectory of arteries in real time, allowing practitioners to choose the best plane for injection.

#### Forehead

The goal of filling the forehead region is to improve the appearance of horizontal wrinkles and to create a more rounded forehead, which is often desired for aesthetic purposes. A classic technique involves deep (supraperiosteal) injections [14]. A horizontal line can be drawn 2 cm above the superior orbit, and the filling can be conducted above this line, with the cannula entering laterally through the temporal crest region [23]. This area is theoretically safer, as the vessels tend to be more superficial. Typically, the entire region is filled with low-elasticity (low G′) hyaluronic acid, which should be gently massaged after the procedure. In some cases, it may be desirable to fill the horizontal and vertical lines of this region more superficially. This is a riskier procedure, as the vessels are also present in this plane.

For ultrasound-guided filling, we apply the same precautions as in the glabella, combining the techniques of “scan before injecting” and “scan while injecting” in the same procedure (Figure 10). Before starting the procedure, the supraorbital and supratrochlear arteries are scanned and marked on the skin (“scan before injecting”). Then, after each insertion of the cannula, it is checked again with the ultrasound to ensure it is not in contact with any vessels (“scan while injecting”).

#### Supraorbital Region

Filling the supraorbital region usually aims to restore the volume of lost fat in this area, gently lifting the eyebrows. For guided filling of this area, we recommend identifying the supraorbital foramen, its corresponding artery, and the plane of injection, which is the Retro-Orbicularis Oculi Fat compartment (ROOF) (Figure 11) before the procedure and monitoring the position of the cannula during treatment. As with other areas of the forehead, we combine the techniques of “scan before injecting” and “scan while injecting”.

A summary of the upper third regions, including the ultrasound-guided injection techniques is presented (Table 1).

## 4. Discussion

Traditional filler injection techniques without the use of ultrasound rely on anatomical landmarks and palpation to guide the placement of fillers. Vascular complications, while relatively rare, can lead to devastating consequences such as vascular occlusion, tissue necrosis, alopecia, and blindness. The introduction of ultrasound-guided techniques with real-time visualization of vascular structures offers an advantage in terms of safety and precision. By visualizing the cannula position relative to blood vessels in real time, practitioners can adjust their approach to maintain a safe distance from critical structures.

Several prior studies describe techniques using ultrasound for upper face injections. Desyatnikova [6] described ultrasound-guided injections in the interfascial plane for the temple area. Most of the previous studies utilize pre-injection mapping including that of Kadouch et al. [7] who also describe the use of ultrasound while injecting lipofilling in the temple. This method provides valuable information and has been shown to increase procedural safety in our experience.

For temporal volumization, our group recommends scanning the region before injection (scan before injecting) for the supraperiosteal technique and scanning during injection (scan while injecting) for subcutaneous or interfascial injections. The “one up, one over” point described by Swift [24] has been re-evaluated, revealing a higher-than-expected incidence of the deep temporal artery at this location with the increased facial ultrasonography use [25]. Real-time ultrasound guidance allows for immediate identification and avoidance of this artery, thereby enhancing procedural safety.

Avoiding arterial embolization is crucial, but it is equally important to consider the presence and course of veins. During temple augmentation, it is important to avoid the medial zygomaticotemporal vein and the middle temporal vein. Real-time ultrasound guidance provides a dynamic view of these veins, allowing for precise navigation and reducing the risk of venous complications.

Identifying a vascular structure during ultrasound-guided injection close to or in contact with the cannula should prompt repositioning the cannula away from the vessel. The minimum safe distance at which the cannula should be from the vessel before injection has not yet been established, but we suggest that this distance should be 1–2 mm. It is also advisable to visualize the needle or cannula tip and the cannula opening, which is usually located away from the tip, and sometimes can be helpful to turn the probe 90 degrees to visualize the vessel in the orthogonal plane and ensure the cannula is not in the vessel.

Real-time ultrasound guidance in the forehead and glabella allows practitioners to adjust the cannula’s plane as needed, minimizing the risk of vascular events by avoiding critical arteries.

The literature presents conflicting descriptions of the forehead ultrasound anatomy, including the frontalis muscle. The frontalis muscle is as a deep, hypoechoic band that can be juxtaposed to the bone [11,26,27,28,29]; however, some authors refer to the subcutaneous fat tissue as the frontalis muscle [14,15,30,31,32]. Tracking the hypoechoic subcutaneous fat tissue with its tortuous septae from the temporal region to the forehead aids in identifying the layers effectively. Additionally, initiating the scan from the lower part of the forehead, where the frontalis muscle connects to the corrugator, might assist the operator in recognizing the muscle more clearly. Dynamic evaluation can also assist in locating the frontalis, as the muscle has a single belly that contracts as a single unit [29]. The rounded structures seen superficially to the muscle at maximum contraction are the condensed subcutaneous fat tissue areas (Figure 6), which some authors interpret as the contracted muscle.

Previous studies described the pathway and depth of the arteries of the forehead and glabella [1,12,13]; however, during ultrasound-guided injection in these areas we notice that the cannula can easily change planes due to the convexity of the forehead. Therefore, ultrasound with real-time visualization of the cannula and blood vessels can improve the safety and efficacy of these injections.

Real-time ultrasound guidance enhances the safety and efficacy of filler injections in the upper face by allowing practitioners to visualize and avoid critical vascular structures. We recommend the adoption of these techniques to reduce the risk of vascular complications. Future research should focus on establishing standardized guidelines for safe injection distances and further elucidating anatomical variations.

The described techniques have some drawbacks. They require specialized equipment and training. The learning curve associated with mastering real-time ultrasound imaging can be steep, potentially limiting widespread adoption. Furthermore, the availability and cost of ultrasound equipment may be prohibitive for some practitioners.

## 5. Conclusions

In conclusion, ultrasound-guided techniques for filler injections in the upper face promise significant improvements in procedural safety and efficacy. By incorporating real-time imaging, practitioners can navigate complex vascular anatomy more effectively, reducing the risk of complications. Ultrasound-guided injections in the upper third of the face underscore the importance of ongoing research and education to optimize these techniques and improve patient outcomes.

## Figures and Tables

**Figure 1 diagnostics-14-01718-f001:**
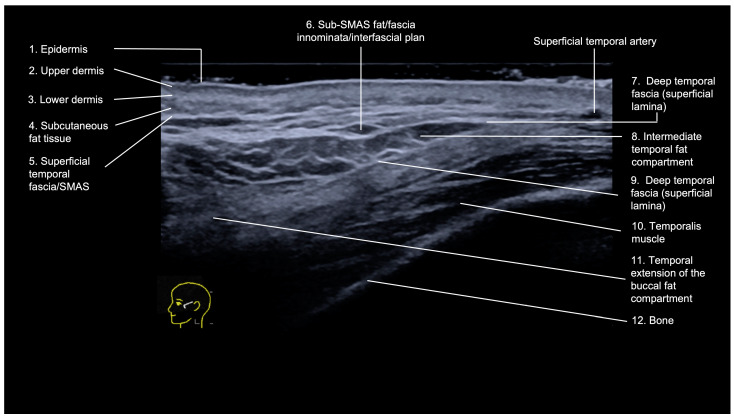
US (greyscale) of the temple showing its layers on the transverse view with an 18 MHz linear probe (ACUSON Sequoia Siemens).

**Figure 2 diagnostics-14-01718-f002:**
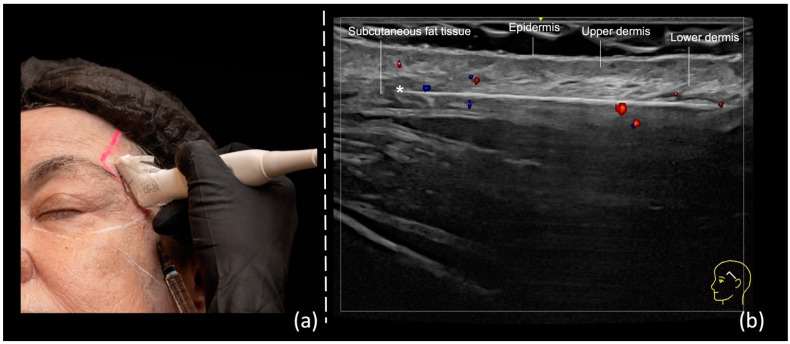
US-guided injection in the temporal region: (**a**) In-plane technique, “scan while injecting” in the subdermis; (**b**) color Doppler US imaging showing cannula parallel to the 24 MHz probe (LOGIQ E10 GE), and its tip (*) outside of vessels (in red or blue).

**Figure 3 diagnostics-14-01718-f003:**
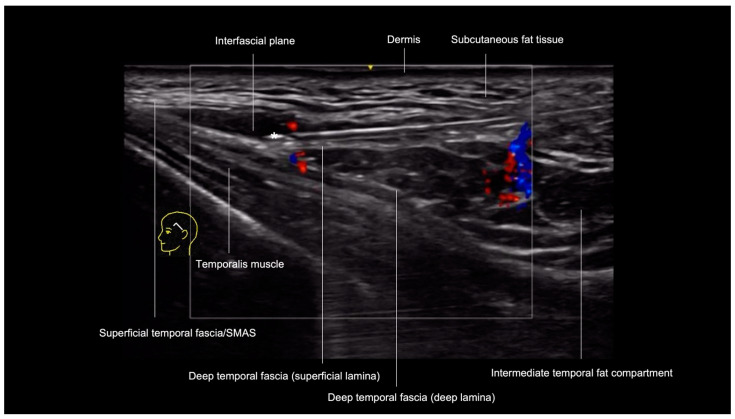
US-guided filler injection in the interfascial plane of the temporal region: color Doppler US imaging with a 20 MHz linear probe (Venue Fit GE) showing the tip of the cannula (*) in the interfascial plane between the superficial temporal fascia and the superficial lamina of the deep temporal fascia. The red/blue color underneath the cannula is an artifact in this case.

**Figure 4 diagnostics-14-01718-f004:**
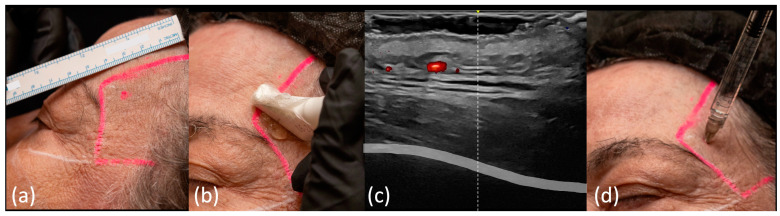
US-guided injection in the deep temporal region: (**a**) The entry point is marked on the skin; (**b**) “scan before injecting” technique, the middle of the probe is positioned at the marker; (**c**) color Doppler US imaging (LOGIQ E10 GE) showing the middle of the screen (yellow arrowhead), which corresponds to the middle of the probe, and is projected (white dotted line) on the deep portion of the temporalis muscle and the periosteum (grey band), where the deep temporal arteries are expected to run. This projection should be extensively checked for the deep temporal artery; the frontal branch of the superficial temporal artery shows in red; (**d**) low supraperiosteal injection is being conducted after US scanning.

**Figure 5 diagnostics-14-01718-f005:**
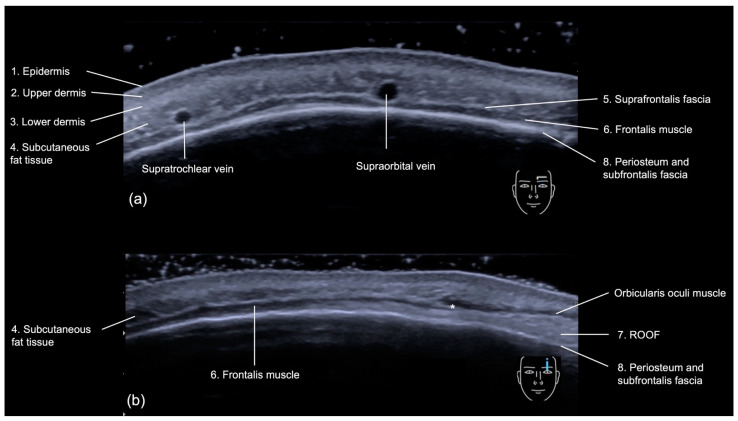
US (greyscale) of the forehead showing its layers: (**a**) transverse view and (**b**) longitudinal view with 18 MHz linear probe (ACUSON Sequoia Siemens), showing the insertion of the frontalis in the orbicularis oculi muscle (*).

**Figure 6 diagnostics-14-01718-f006:**
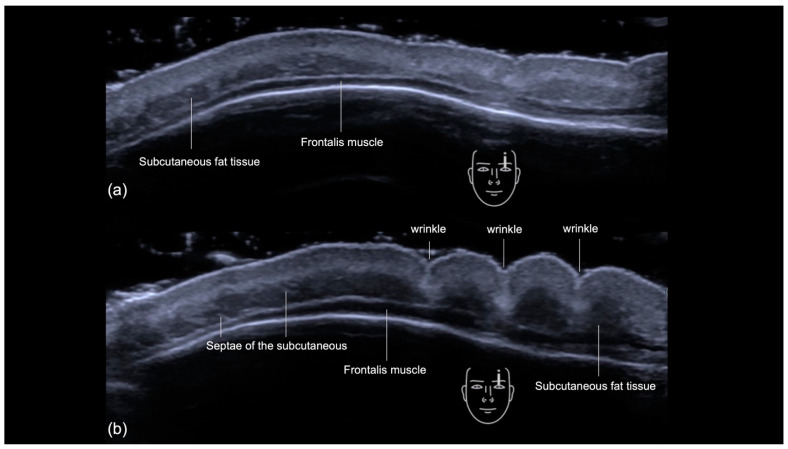
US (greyscale) demonstrating the contraction of the forehead (transverse view, 18 MHz probe ACUSON Sequoia Siemens): (**a**) patient at rest, (**b**) patient frowning, showing the uniform contraction of the frontalis muscle as a single unit. Notice the increase in thickness of the frontalis muscle and the condensation of subcutaneous fat tissue as the frontal lines (wrinkles) are formed.

**Figure 7 diagnostics-14-01718-f007:**
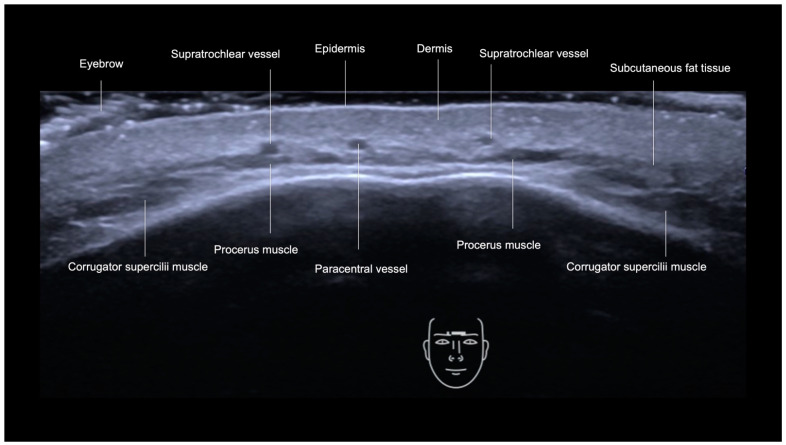
US (greyscale) of the glabella, transverse view with an 18 MHz probe (ACUSON Sequoia Siemens), demonstrating its layers.

**Figure 8 diagnostics-14-01718-f008:**
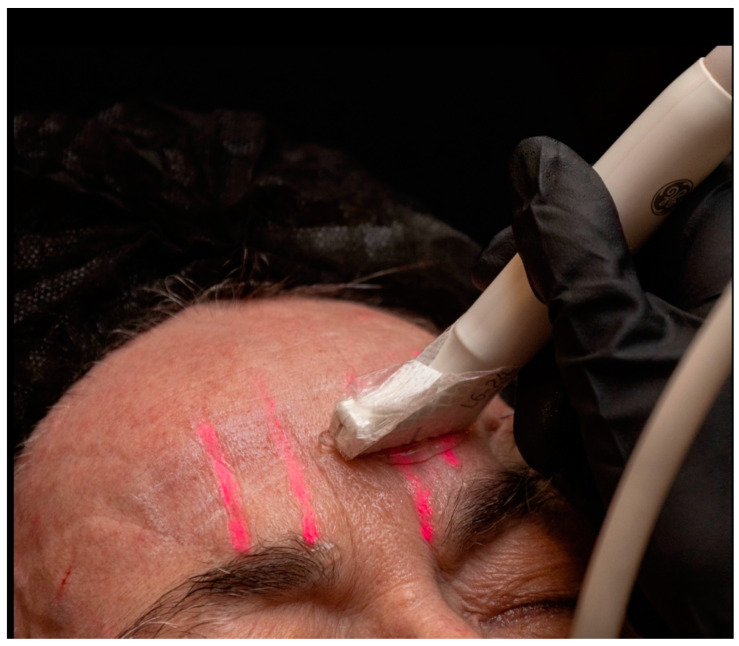
Supratrochlear and supraorbital arteries being marked on the skin as DUS is performed in transverse plane of the artery.

**Figure 9 diagnostics-14-01718-f009:**
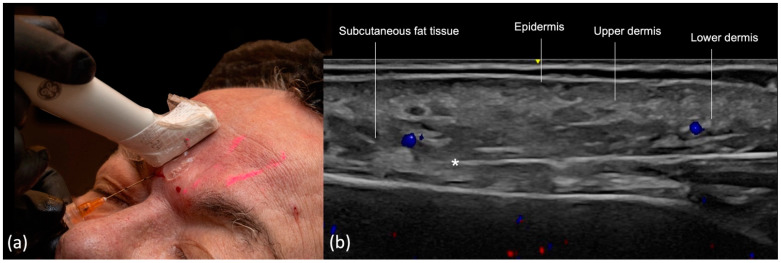
US-guided injection in the glabella. The cannula is placed after the region was scanned and the arteries marked (Scan before injecting). (**a**) Scan while injecting, (**b**) color Doppler US imaging (LOGIQ E10 GE) is used to ensure the cannula (*) is outside of vessels (in blue).

**Figure 10 diagnostics-14-01718-f010:**
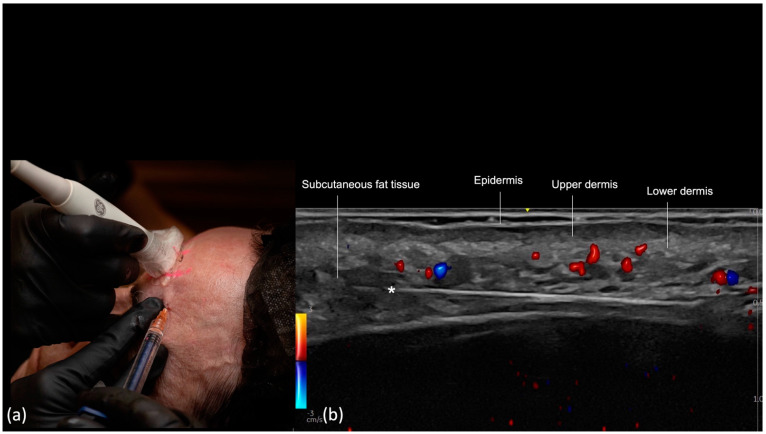
US−guided injection in the Forehead. The cannula is placed after the region was scanned and the arteries marked (Scan before injecting). (**a**) Scan while injecting, (**b**) color Doppler US imaging (LOGIQ E10 GE) is used to ensure the cannula (*) is outside of vessels (in red or blue). Notice the tip of the cannula pointing upwards due to the convexity of the frontal bone.

**Figure 11 diagnostics-14-01718-f011:**
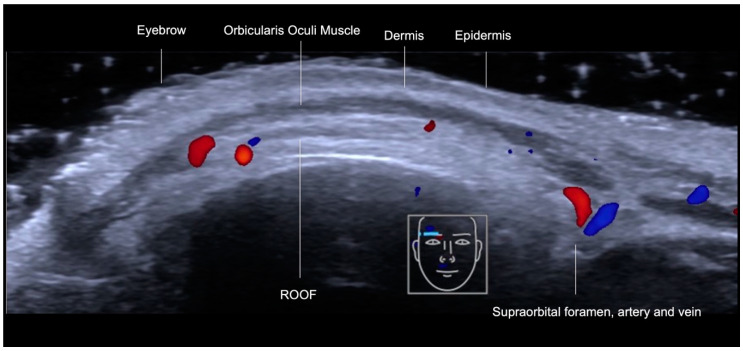
Color Doppler US showing a transverse view of the supraorbital foramen and artery in red and vein in blue, and the Retro-Orbicularis Oculi Fat (ROOF), underneath the orbicularis oculi muscle with an 18 MHz probe (ACUSON Sequoia Siemens).

**Table 1 diagnostics-14-01718-t001:** Summary of the upper third regions and ultrasound-guided injection techniques.

Region	Scan before Injecting	Scan While Injecting	Desired Planes to Inject	What to Avoid in This Region
Temporal Region Technique 1: Subdermal filler placement	No	Yes	Subcutaneous fat tissue	Superficial temporal artery and vein
Temporal Region Technique 2: Interfascial filler placement	No	Yes	Interfascial plane between the superficial temporal fascia and deep temporal fascia (superficial lamina)	Superficial temporal artery and vein, middle temporal vein, sentinel vein
Temporal Region Technique 3: Low supraperiosteal filler placement (“one up, one over” technique)	Yes	No	Supraperiosteal plane	Anterior deep temporal artery
Frontal Region: Glabella	Yes	Yes	Usually superficial, may vary according to the depth of the arteries on DUS	Supratrochlear, central and paracentral arteries
Frontal Region: Forehead	Yes	Yes	Usually deep, may vary according to the depth of the arteries on DUS	Supraorbital, supratrochlear, central and paracentral arteries
Frontal Region: Supraorbital region	Yes	Yes	ROOF	Supraorbital artery

## Data Availability

Data sharing is not applicable to this article.

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
