# Peer review of "Best Practices for the Use of High-Frequency Ultrasound to Guide Aesthetic Filler Injections—Part 1: Upper Third of the Face"

_diagnostics, 2024, doi:10.3390/diagnostics14161718_

Round 1

Reviewer 1 Report

Comments and Suggestions for Authors

·        Methodology: This is not a study but more a kind of guideline for those who use or want to integrate ultrasound into filler treatments. The set up should be different; there are not methods / results ect

·        Temple: if needed, the supraperiosteal filler placement can also be done US- guided with an angle of the needle instead of perpendicular, the result will be the same.

·        In the Legenda of the images,  the type of devices should be mentioned as well

·        If possible, it would be nice to show in image 4c an image also showing the temporal bony fossa where you expect the vessels in the one up one over

·        Image 5: is the suprafrontalis fascia pointed out correctly? It might be the clarity of the image but the fascia is not so sharp

·        Forehead: could you define “Flexible” HA filler in rheology / G prima etc

·        Discussion and conclusions: “The clinical implications of our findings are significant” : it is more a guideline; there are no specific findings mentioned in this article.

Reviewer 2 Report

Comments and Suggestions for Authors

The detailed exploration of the ultrasonographic anatomy of the temples, forehead, and glabella, coupled with reproducible injection techniques, provides a valuable resource for practitioners.This paper can undoubtedly serve as a guide for aesthetic professionals, enhancing patient outcomes and minimizing risks. 

Congratulations to the authors!
